# Disruption of Cytosolic Folate Integrity Aggravates Resistance to Epidermal Growth Factor Receptor Tyrosine Kinase Inhibitors and Modulates Metastatic Properties in Non-Small-Cell Lung Cancer Cells

**DOI:** 10.3390/ijms22168838

**Published:** 2021-08-17

**Authors:** Po-Wen Shen, Chun-Te Ho, Shih-Hsin Hsiao, Yu-Ting Chou, Yi-Cheng Chang, Jun-Jen Liu

**Affiliations:** 1Program in Molecular Medicine, National Yang Ming Chiao Tung University and Academia Sinica, Taipei 112, Taiwan; janothan1983@gmail.com; 2Institute of Biomedical Sciences, Academia Sinica, Taipei 115, Taiwan; dtmed200@gate.sinica.edu.tw; 3Graduate Institute of Medical Sciences, College of Medicine, Taipei Medical University, Taipei 110, Taiwan; homenjer@gmail.com; 4Division of Pulmonary Medicine, Department of Internal Medicine, Taipei Medical University Hospital, Taipei 110, Taiwan; hsiaomd@tmu.edu.tw; 5Department of Life Science, National Tsing Hua University, Hsinchu 300, Taiwan; ytchou@life.nthu.edu.tw; 6Department of Internal Medicine, National Taiwan University Hospital, Taipei 110, Taiwan; 7Graduate Institute of Medical Genomics and Proteomics, National Taiwan University, Taipei 110, Taiwan; 8School of Medical Laboratory Science and Biotechnology, Taipei Medical University, Taipei 110, Taiwan; 9Ph.D. Program in Medical Biotechnology, Taipei Medical University, Taipei 110, Taiwan; 10Ph.D. Program in Biotechnology Research and Development, Taipei Medical University, Taipei 110, Taiwan

**Keywords:** folate integrity, folate deprivation, EGFR-TKI, NSCLC, resistance acquisition, promoted invasiveness

## Abstract

Patients with advanced-stage non-small-cell lung cancer (NSCLC) are susceptible to malnutrition and develop folate deficiency (FD). We previously found that folate deprivation induces drug resistance in hepatocellular carcinoma; here, we assessed whether disrupted cytoplasmic folate metabolism could mimic FD-induced metastasis and affect the sensitivity of NSCLC cells to epidermal growth factor receptor tyrosine kinase inhibitors (EGFR-TKIs). We examined whether cytosolic folate metabolism in NSCLC cells was disrupted by FD or the folate metabolism blocker pemetrexed for 1–4 weeks. Our results revealed an increase in NF-κB overexpression–mediated epithelial-mesenchymal transition biomarkers: N-cadherin, vimentin, matrix metalloproteinases (MMPs), SOX9, and SLUG. This finding suggests that the disruption of folate metabolism can drastically enhance the metastatic properties of NSCLC cells. Cytosolic FD also affected EGFR-TKI cytotoxicity toward NSCLC cells. Because SLUG and N-cadherin are resistance effectors against gefitinib, the effects of SLUG knockdown in folate antagonist–treated CL1-0 cells were evaluated. SLUG knockdown prevented SLUG/NF-κB/SOX9-mediated invasiveness and erlotinib resistance acquisition and significantly reduced pemetrexed-induced gelatinase activity and MMP gene expression. To summarize, our data reveal two unprecedented adverse effects of folate metabolism disruption in NSCLC cells. Thus, the folic acid status of patients with NSCLC under treatment can considerably influence their prognosis.

## 1. Introduction

Lung cancer is the leading cause of cancer-related death worldwide [1,2]. Traditional first- or second-line chemotherapy strategies for wild-type epidermal growth factor receptor (EGFR)–expressing non-small-cell lung cancer (NSCLC) include cisplatin- or carboplatin-based combinations; other agents, such as paclitaxel, docetaxel, and gemcitabine, are considered second-line treatments [3,4,5,6]. For chemoresistant patients, third-line treatments targeting the EGFR pathway, such as gefitinib or erlotinib, are considered [6]. Although the clinical use of EGFR tyrosine kinase inhibitors (EGFR-TKIs) may improve the prognosis of patients with NSCLC, most patients exhibit resistance to EGFR-TKIs due either to genetic endowments [7] or acquired resistance after long-term chemotherapy. Therefore, identifying the potential causes of resistance to EGFR-TKIs is critical.

Malnutrition is common among patients with cancer but is often ignored [8]. As many as 31–97% of patients with cancer have malnutrition [9,10,11,12]. Although malnutrition is relatively uncommon in patients with lung cancer, Xara and colleagues reported that 35.7% of patients with NSCLC are malnourished [13]. Patients with advanced-stage NSCLC can also have malnutrition, which is associated with poor prognosis [14]. Accordingly, the Nutritional Risk Index can help identify patients with early-stage lung cancer who are at risk of postoperative complications [15].

Lower serum folate concentrations were reported to be associated with a higher incidence of lung cancer [14]. Because folate assists in the formation of S-adenosylmethionine (SAM), which affects downstream gene transcription, its deficiency can deplete cellular SAM and subsequently induce proto-oncogene expression [15,16]. Folate deficiency increases uracil misincorporation into DNA; persistent folate deficiency leads to DNA damage and chromosomal disruption, resulting in increased cell mutation and cancer [17,18,19]. Thus, malnutrition, including folate deprivation (FD), in patients with advanced NSCLC may be associated with a metastatic phenotype related to hypomethylation-mediated tumorigenesis [3,4,5] and cell shape remodeling [20].

FD can enhance invasion [21], epithelial-mesenchymal transition (EMT) [22], and chemoresistance [23] in different cancer cells. The mutation of an enzyme responsible for folate metabolism in cancer cells can alter cytosolic folate integrity and modulate the response of folate inhibitors [24]. Therefore, the disruption of folate metabolism can increase cancer heterogeneity, reduce sensitivity to immunotherapy [25], and promote some molecular mechanisms underlying malignancy. Chang et al. [26] identified the EMT-modulating SLUG molecule as a contributor to EGFR-TKI resistance. Defects in folate metabolism can also enhance colon cancer cell invasiveness and trigger multidrug resistance in hepatoma cells [18,20]. The current study assessed the relationship between cytosolic folate integrity disruption and invasiveness/resistance to therapy in NSCLC cells.

## 2. Results

### 2.1. Disruption of Cytosolic Folate Integrity Induces EMT Transformation in NSCLC Cells

To identify the effect of cytosolic folate disruption in NSCLC cells, three EGFR wild-type NSCLC cell lines—CL1-0, CL1-5, and A549—were cultured in FD medium or treated with the folate antagonist pemetrexed. Morphologically, CL1-0 and A549 cells, but not the CL1-5 subline, exhibited cuboidal contours held together in tight layers by cell-cell adhesion and a loss of motility. However, when these cells were cultured in FD medium or treated with 30 ng/mL pemetrexed, all three cell types underwent drastic morphological changes after as little as 1 week and developed elongated spindle shapes and a motile phenotype (Figure 1A). To determine which type of transformation occurred, we identified the emerging acquired biomarkers of EMT. The disruption of folate integrity induced E-cadherin downregulation in CL1-0 and CL1-5 cells but not in A549 cells; additionally, it upregulated N-cadherin or vimentin in all three NSCLC cell lines (Figure 1B,C). Similarly, EMT of NSCLC cells by folate disruption caused NF-κB upregulation (Figure 1D) [27].

### 2.2. Disruption of Cytosolic Folate Integrity Promotes Increased NSCLC Cell Invasiveness by Upregulating Matrix Metalloproteinase Activities

To further examine the effect of cytosolic folate disruption on EMT transformation in NSCLC cells, we analyzed NSCLC cell invasiveness. The cells were cultured under FD or treated with pemetrexed. The invasiveness of NSCLC after 2–3 weeks of treatment was measure by the Matrigel method (Figure 2A). Compared with the relatively normal culture control cells, at the end of the 3-week pemetrexed treatment, these cell lines exhibited magnitudes of migration ability changes (expressed as folds) of 12.1 ± 4.1 (CL1-0), 5.2 ± 3.3 (CL1-5), and 27.2 ± 11.0 (A549) (Figure 2B). Consistent results were obtained for all three NSCLC cell lines after FD treatment: fold change of 10.5 ± 4.4 (CL1-0), 2.9 ± 0.2 (CL1-5), and 9.05 ± 2.9 (A549) (Figure 2C). After FD or pemetrexed treatment, matrix metalloproteinases (MMPs) 1, 2, 7, and 9 were induced in CL1-0 and CL1-5 cell lines, as determined through real-time quantitative reverse transcription-polymerase chain reaction (RT-qPCR) or evaluation of their gelatinase activity; these findings correlated with their increased invasiveness (Figure 2D–F). Together, these results revealed that the antifolate environment mediated EMT and invasiveness.

### 2.3. Folate Antagonist Enhances EGFR-TKI Resistance in NSCLC Cells

EMT transformation of NSCLC cells is associated with acquired resistance to EGFR inhibitors [28,29,30]. Therefore, we identified the effect of cellular folate integrity on the sensitivity of EGFR-TKIs. Both CL1-0 and CL1-5 cell lines were cultured in mediums without or with 30 ng/mL pemetrexed for 2–3 weeks, followed by the addition of EGFR-TKIs at various concentrations (gefitinib or erlotinib; 0.1–100 μM). The half-maximal inhibitory concentration (IC_50_, μM) of both EGFR-TKIs for these cells was then calculated. The basal IC_50_ values for CL1-0 and CL1-5 cells with gefitinib were 4.7 ± 1.4 and 13.1 ± 4.2 μM, respectively, and those of CL1-0 and CL1-5 cells with erlotinib were 2.7 ± 0.1 and 2.6 ± 1.0 μM, respectively, indicating higher sensitivity to erlotinib.

Pemetrexed treatment rendered both CL1-0 or CL1-5 cells highly resistant to gefitinib and erlotinib, as reflected by the drastic increases in IC_50_ values to 27.2 ± 2.0 and 32.1 ± 6.2 μM, respectively, with gefitinib and 33.8 ± 2.1 and 51.9 ± 12.2 μM, respectively, with erlotinib (Figure 3A,B). These data demonstrate that disruption of folate metabolism can promote EGFR-TKI resistance acquisition in NSCLC cells.

### 2.4. Acquisition of FD-Mediated EGFR-TKI Resistance in NSCLC Cells Is Mechanistically Linked to SLUG and N-Cadherin Upregulation

Because the E-cadherin–regulating transcription factors SLUG and N-cadherin promote gefitinib resistance [26,31], we investigated the link between folate disruption and the acquisition of drug resistance in NSCLC cells. Using RT-qPCR, we found that both pemetrexed and FD reduced E-cadherin expression and increased SLUG expression (Figure 4A,B). Using Western blotting, we confirmed that pemetrexed increases nuclear expression of SLUG and N-cadherin in these cells (Figure 1B and Figure 4C).

### 2.5. SLUG Silencing Suppresses Invasiveness and Mitigates Erlotinib Resistance Acquisition in NSCLC Cells

To demonstrate the correlation between invasiveness and EGFR-TKI resistance with SLUG protein induction, we performed SLUG knockdown by using the lentivirus delivery technique. As shown in Figure 5A, SLUG expression in CL1-0 knockdown cells was significantly lower than in scramble cells under the same pemetrexed-treated conditions. Moreover, SLUG silencing significantly reduced N-cadherin expression (Figure 5A). SLUG protein knockdown also resulted in fewer invaded cells than the numbers of control and scramble si-RNA groups (Figure 5B). These data correlated with MMP gene expression (MMP1, MMP2, MMP7, and MMP9) and gelatinase activity after SLUG silencing (Figure 5C,D). We also demonstrated that SLUG silencing of CL1-0 cells reduced the acquisition of resistance to erlotinib, with IC_50_ values dropping from 34.6 ± 0.9 (FD control) to 1.3 ± 0.5 μM (si-SLUG) (Figure 5E).

### 2.6. EMT in Disrupted Folate Metabolism Is Mechanistically Linked to NF-κB/Sox9 Pathway Activation

In addition to SLUG overexpression, mild pemetrexed treatment activated Sox9, a downstream target gene of the sonic hedgehog (Shh) pathway triggered by cross-reaction with NF-κB [21] (Figure 1C and Figure 6A,B). To further confirm the relationship between Sox9 and NF-κB expressions, we transfected pemetrexed-treated CL1-0 cells with shRNA (si-RELA-212 and si-RELA-689) by using a lentiviral vector. RT-qPCR indicated that Sox9 expression was severely reduced in NF-κB-knockdown CL1-0 cells, implying that pemetrexed-mediated Sox9 activation was controlled by NF-κB expression (Figure 6C). Notably, this suggests that SLUG might be involved in NF-κB activation and expression after pemetrexed treatment. We confirmed that SLUG knockdown (si-SLUG-654 and si-SLUG-871) significantly reduced both NF-κB activation and Sox9 expression (Figure 6D,E).

## 3. Discussion

Lung cancer remains a major cause of cancer death worldwide [1]. The development of EGFR-TKIs [32] has raised hopes of improved prognosis in patients with NSCLC; however, the effectiveness of EGFR-TKI monotherapy is challenged by the acquisition of resistance to EGFR-TKIs due to intrinsic genetic factors or extrinsic long-term chemotherapy. Therefore, identifying the underlying mechanisms promoting resistance to EGFR-TKIs in NSCLC is essential. Public health reports have indicated that lung cancer mortality is generally associated with malnutrition in advanced stages [3], and FD is a risk factor for colon cancer and liver cancer [21,23]. However, its effects on patients with NSCLC and treatment protocols remain underexplored. In this study, we clarified whether the disruption of folate integrity could synergistically assist EGFR-TKIs in eradicating NSCLC cells. After culturing CL1-0 and CL1-5 NSCLC cells in FD mediums or with low doses of pemetrexed (30 ng/mL) for 2–3 weeks, we observed a dramatic alteration in cell morphology. Compared with control cells with round and flat epithelial-like contours, cells with lower folate levels became mesenchymal (fibroblastic-like) and phenotypically amoeboid, indicating EMT. Furthermore, the disruption of folate integrity in NSCLC cells can also lead to the overexpression of EMT biomarkers, including SLUG and N-cadherin, a pair of newly discovered effectors that cause resistance to gefitinib [26,31]. SLUG, a zinc-finger transcription factor of the Snail superfamily, is the crucial EMT regulator responsible for conferring acquired resistance to EGFR-TKIs in NSCLC cells. It promotes tumor cell invasiveness through increased MMP-2 activity and E-cadherin suppression [24]. This E-cadherin downregulation and N-cadherin upregulation (known as cadherin switch) occur in cancers of epithelial origin when the tumor advances to a more malignant phenotype. This evidence is consistent with our MMPs and cell migration data after SLUG silencing (Figure 5A,C,D).

Chang et al. demonstrated that the EMT regulator SLUG also contributes to the development of resistance to gefitinib in patients with lung adenocarcinoma containing EGFR-activating mutations [26]. Consistently, our data indicate that pemetrexed-mediated upregulation of SLUG expression can lead to acquired resistance against erlotinib and gefitinib (Figure 3 and Figure 5E). Therefore, the development of SLUG-targeted drugs could be a promising strategy to enhance the effectiveness of lung cancer therapy. Yamauchi et al. stated that cadherin switch could also contribute, at least in part, to the survival mechanisms of gefitinib-resistant NSCLC cells, suggesting that under the influence of pemetrexed, N-cadherin expression can promote the acquisition of resistance to EGF-TKIs; this is consistent with our findings [31]. We also discovered that the transcription factor Sox9, a member of the SOX gene superfamily, was concomitantly overexpressed in all treated NSCLC cells and regulated by SLUG (Figure 6). In parallel with these observations, we found that pemetrexed-provoked NF-κB overexpression regulates these phenomena [27].

Pemetrexed-induced metabolic stress may trigger the activation of the Shh pathway, similar to FD, because Sox9 is the downstream target gene of this pathway [21,33]. Like other cancer cells, Sox9 and SLUG enhance the metastatic propensity of NSCLC cells. [34]. Pemetrexed-induced simultaneous SLUG and Sox9 overexpression in NSCLC cells are correlated with increased EMT and an enhanced propensity for metastasis [34], as reflected in our invasion assay data (Figure 2 and Figure 6). Although these findings indicate that pemetrexed plays a crucial role in inducing increased metastatic propensity by promoting EMT induction in NSCLC cells, the mechanisms underlying the aberrant expression of EMT biomarkers in NSCLC cells under the influence of folate metabolism disruption remain unclear.

In addition to its ability to disrupt folate-dependent metabolism, pemetrexed can trigger metabolic stress by mimicking FD-induced homocysteine (Hcy) accumulation by inhibiting its interconversion back to methionine. Hcy accumulation leads to hydrogen peroxide–mediated and iNOS-mediated oxidative/nitrosative stress and apoptotic cell death in human hepatoma Hep G2 cells [32,33,35,36,37,38,39,40]. Similarly, Wang et al. reported that FD could enhance the invasiveness of colon cancer cells via Shh signaling through promoter hypomethylation and interactions with the NF-κB pathway [21]. This finding is consistent with the notion that FD-induced oxidative stress may promote redox adaptation, enabling cancer cells to survive under increased ROS stress and contributing to enhanced metastatic propensity [41,42,43]. Thus, pemetrexed-induced Hcy accumulation and oxidative stress can together lead to redox adaptation, resulting in EMT. Consequently, the dual overexpression of SLUG and Sox9 could contribute to increased metastatic propensity while SLUG and N-cadherin confer resistance to erlotinib and gefitinib in these NSCLC cells.

In conclusion, cytosolic folate disruption through both FD and treatment with the multitargeted folate antagonist pemetrexed can unexpectedly trigger two distinct adverse effects on NSCLC cells. First, cytosolic folate metabolic stress can promote metastatic propensity through EMT induction and the dual overexpression of SLUG and Sox9. Second, SLUG and N-cadherin overexpression induced by folate metabolic stress can contribute to EGFR-TKI resistance. A summary of the pathways involved in the disruption of folate metabolism, leading to increases in metastatic propensity and acquisition of EGFR-TKI resistance, is presented in Figure 7. Our results suggest that treating NSCLC through the disruption of folate metabolism requires serious consideration and precautions.

## 4. Materials and Methods

### 4.1. Cell Lines, Culture Mediums, Drugs, and Antibodies

The lung adenocarcinoma cell lines CL1-0, CL1-5, and A549 were provided by Dr. Cheng-Wen Wu and Dr. Pan-Chyr Yang [44,45,46]. Cells were grown in RPMI 1640 medium with 10% heat-inactivated fetal bovine serum and cultured at 37 °C in a humidified atmosphere with 5% CO_2_. The folate-deficient medium, RPMI 1640 medium without folic acid, was purchased from Gibco (Thermo Fisher Scientific, Waltham, MA, USA). Fetal bovine serum was dialyzed at 4 °C for 16 h against 10 volumes of sterile phosphate-buffered saline and used under the FD condition to minimize the amount of folate from exogenous sources. Pemetrexed from Lilly (Indianapolis, IN, USA). Gefitinib (Iressa) was obtained from AstraZeneca (Macclesfield, Cheshire, UK), and erlotinib (Tarceva) was obtained from Roche (Basel, Switzerland). Antibodies specific for E-cadherin (H-108), vimentin (V9), SLUG (A-7) (Santa Cruz Biotechnology, Dallas, TX, USA), N-cadherin (22200002) (Novus International, Saint Charles, MO, USA), Sox9 (EPR14335) (Epitomics, Wembley, London, UK), and actin (AC-15) (Sigma-Aldrich, St. Louis, MO, USA) were used in Western blot analysis.

### 4.2. Cell Viability Assay

Pemetrexed-treated cells seeded in 96-well plates were exposed to different concentrations of EGF-TKI (gefitinib or erlotinib) in triplicate for 72 h. We performed cell viability assays according to the sulforhodamine B (Sigma-Aldrich) method [47]. IC_50_ was determined and indicated 50% growth inhibition compared with the control group.

### 4.3. Western Blot Analysis

Nuclear protein extract was prepared with the Nuclei EZ Prep Kit (NUC101) (Sigma-Aldrich, St. Louis, MO, USA), and total protein extract was prepared with total lysis buffer (Enzo Life Sciences, Lausen, Switzerland). Equal amounts of protein extract were subjected to sodium dodecyl sulfate polyacrylamide gel electrophoresis (SDS-PAGE) and transferred to polyvinylidene difluoride membranes (IPVH00010) (Merck Millipore, Darmstadt, Germany). After being transferred, the membranes were probed with primary antibodies against N-cadherin (1:10^4^), vimentin (1:500), Sox9 (1:1000), SLUG (1:500), and actin (1:10^4^) for 12–24 h. After incubation with horseradish peroxidase-conjugated secondary antibody, the detection was performed through enhanced chemiluminescence (WBULS0500) (Merck Millipore, Darmstadt, Germany). 

### 4.4. RT-qPCR

NSCLC cells (CL1-0, CL1-5, and A549) were lysed with TriZol LS reagent (Thermo Fisher Scientific, Waltham, MA, USA) for RNA purification. We reverse-transcribed RNA by using a RevertAid cDNA Synthesis Kit (K1622, Thermo Fisher Scientific, Waltham, MA, USA). Gene expression assays were performed with the Maxima SYBR Green/ROX qPCR Master Mix (K0221, Thermo Fisher Scientific, Waltham, MA, USA) on a LightCycler 480 System. The primers are listed in Appendix A. Threshold cycle (Ct) values were evaluated with LightCycler 480 Software v1.5.0. Transcript ΔCt levels were normalized according to the 18S gene (endogenous control). The levels of change (ΔΔCt) in the target genes are presented as a multiple of the change in expression (2^ΔΔCt^).

### 4.5. Lentivirus-Mediated Knockdown of NF-κB (RELA) or SLUG Expression

siRNAs targeting the RELA (si-RELA-212: AGGAGCACAGATACCACCAAG ACC and si-RELA-689: GGGATGAGATCTTCCTACT) and SLUG (si-SLUG-654: GCTGTAAATA-CTGTGACAA and si-SLUG-871: GCAGACCCATTCTGATGTA) were cloned into the pLV-H1-EF1a-GFP-Puro vector and packaged in a lentivirus (Biosettia, San Diego, CA, USA). Lentiviruses containing medium and polybrene were added to the cultures of pemetrexed-treated cells and subjected to puromycin selection.

### 4.6. Invasion Assay and Gelatin Zymography

We evaluated cell invasion by following the modified transwell method. Briefly, a PET membrane precoated with 100 μL of Matrigel (2 mg/mL) (Becton, Dickinson and Company, Billerica, MA, USA) was used. 1 × 10^5^ of cells were seeded in the upper chamber of transwell when the matrigel is solidified. After 16–18 h incubation, the invading cells were fixed and stained with Liu’s stain and observed under a microscope (Leica, Germany).

Serum-free conditioned mediums with activated MMP2 and MMP9 were evaluated through gelatin zymography. In brief, conditioned mediums were concentrated in a concentrator (Corning). Equal amounts of protein were then loaded onto 8% polyacrylamide gels incorporated with 0.5% gelatin for electrophoresis and stained with Coomassie brilliant blue R-250 (Sigma-Aldrich, St. Louis, MO, USA). The gelatinolytic activity was identified as a white band against a dark blue background.

### 4.7. Statistical Analysis

Differences at a level of *p* < 0.05 for a paired Student’s *t* test (Sigma plot 8.0, Systat Software Inc., San Jose, CA, USA) were considered statistically significant.

## Figures and Tables

**Figure 1 ijms-22-08838-f001:**
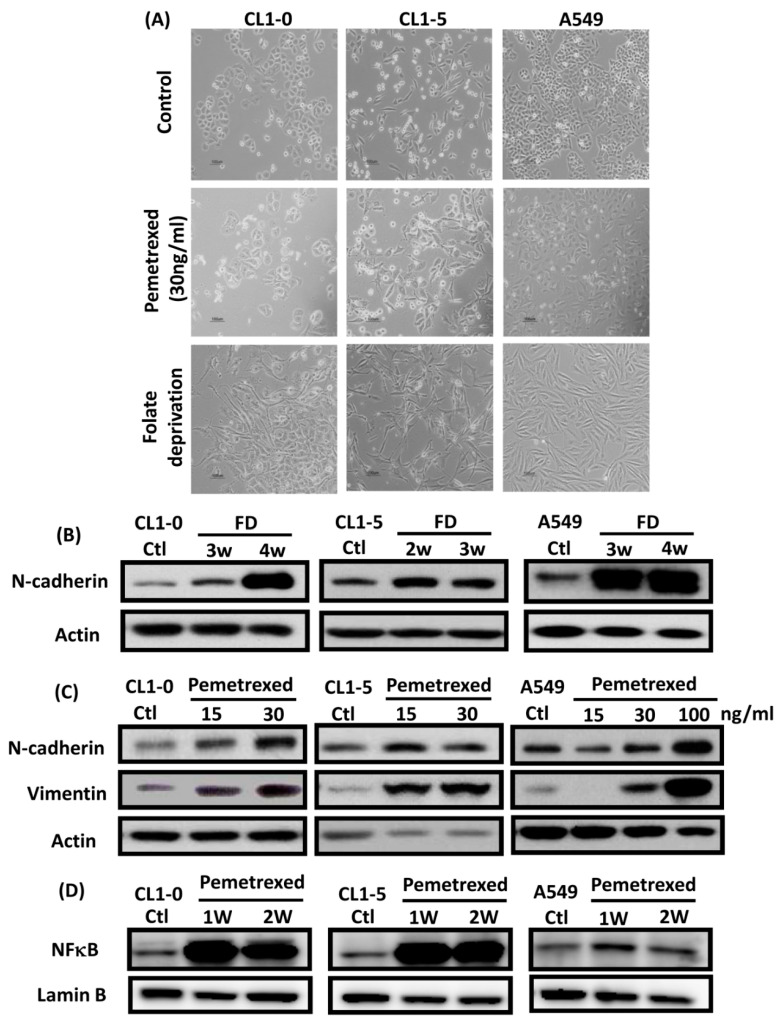
Disruption of cytosolic folate integrity induces EMT in NSCLC cells. (**A**) Morphological images at 1 week (scale bar: 100 μm). (**B**) EMT biomarker N-cadherin increase in NSCLC cell lines after 2–4 weeks of folate-deprived. (**C**) N-cadherin and vimentin Protein expressions increased in NSCLC cell lines after treatment with various concentrations of pemetrexed for 1 week. (**D**) NF-κB expression in NSCLC cells after treatment with pemetrexed for 1 or 2 weeks, as detected through Western blotting.

**Figure 2 ijms-22-08838-f002:**
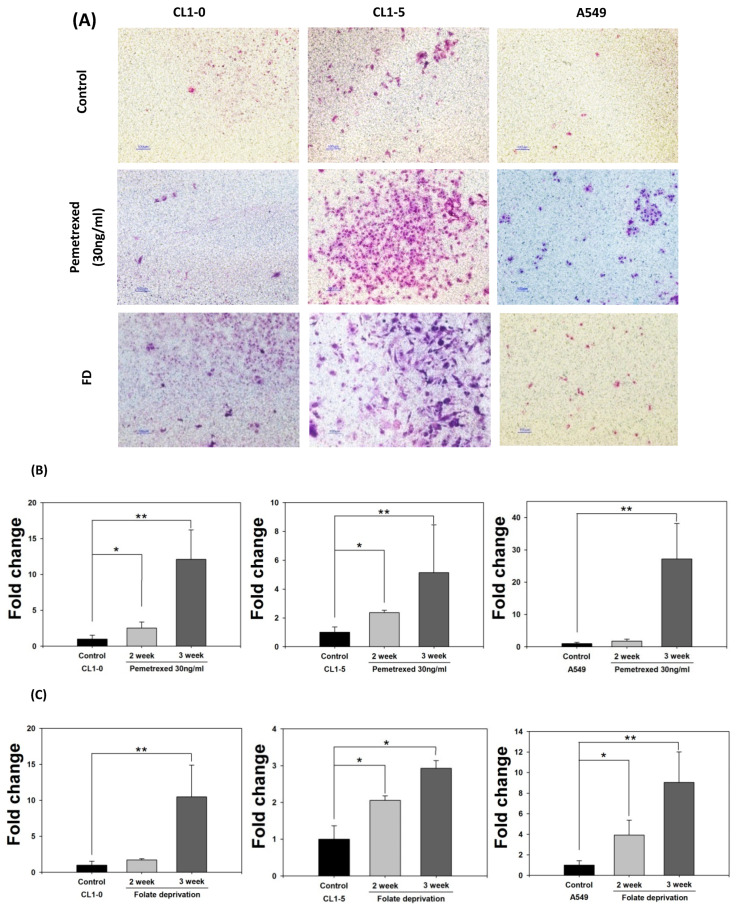
Disruption of cytosolic folate integrity enhances NSCLC cell invasiveness by upregulating MMP activities. (**A**) Typical Matrigel images and NSCLC cell invasiveness with or without disruption of cytosolic folate integrity were compared. Cells that had invaded through the matrix gel were quantified according to the number of stained cells through Liu’s stain method in all fields of each multiwell cell culture insert. The invasive cells were stained purple. Scale bar (blue): 100 μm. (**B**) Magnitudes of migration ability changes (expressed as folds) of pemetrexed-treated NSCLC cell lines. (**C**) Magnitudes of migration ability change (expressed as folds) of folate-deprived NSCLC cell lines. (**D**) The mRNA expressions of MMPs, including MMP1, MMP2, MMP7, and MMP9 induced by pemetrexed (30 ng/mL) for 1, 2, or 3 weeks in CL1-0 or CL1-5 cells were determined through RT-qPCR (black bar: control, gray bar: pemetrexed for 1 week, dark gray bar: pemetrexed for 2 weeks, and light gray bar: pemetrexed for 3 weeks). (**E**) CL1-0 cells under FD treatment for 2, 3, or 4 weeks exhibited increased MMP1, MMP2, and MMP9 mRNA expressions, but no significant increases were observed in folate-deprived CL1-5 cells (black bar: control, gray bar: FD for 2 weeks, dark gray bar: FD for 3 weeks, and light gray bar: FD for 4 weeks). (**F**) Gelatinase activities of folate-deprived or pemetrexed-treated (30 ng/mL) CL1-0 or CL1-5 cells were also analyzed through gelatin-containing zymography. * *p* < 0.05; ** *p* < 0.01; *** *p* < 0.001.

**Figure 3 ijms-22-08838-f003:**
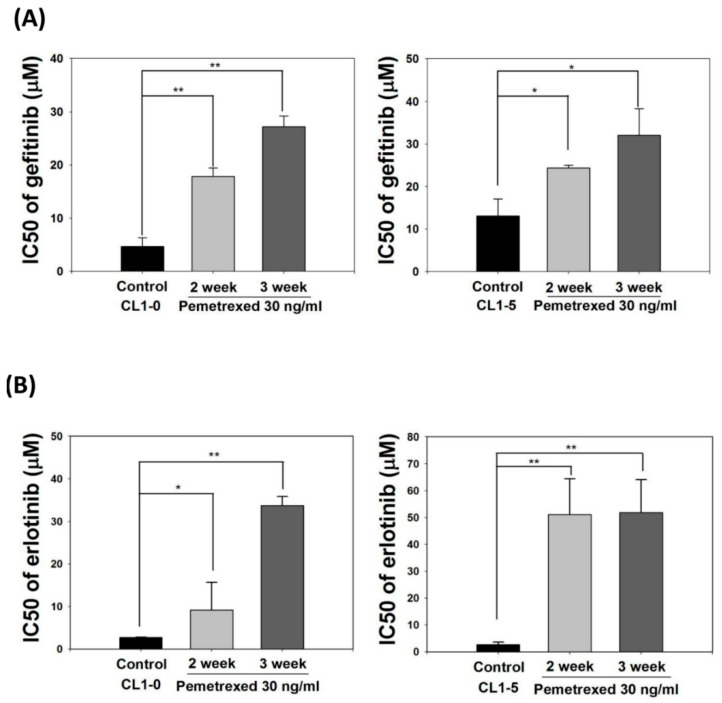
Pemetrexed aggravates EGFR-TKI resistance acquisition in NSCLC cells. NSCLC cells (CL1-0 and CL1-5) were cultured in pemetrexed-free or pemetrexed-containing (30 ng/mL) mediums for 2–3 weeks, followed by treatment with various concentrations of (**A**) gefitinib or (**B**) erlotinib (0.1–100 μM) for an additional 3 days in 96-well plates. IC_50_ values of gefitinib or erlotinib for these NSCLC cells demonstrated that pemetrexed unilaterally promoted EGFR-TKI resistance acquisition in the NSCLC cells. * *p* < 0.05; ** *p* < 0.01.

**Figure 4 ijms-22-08838-f004:**
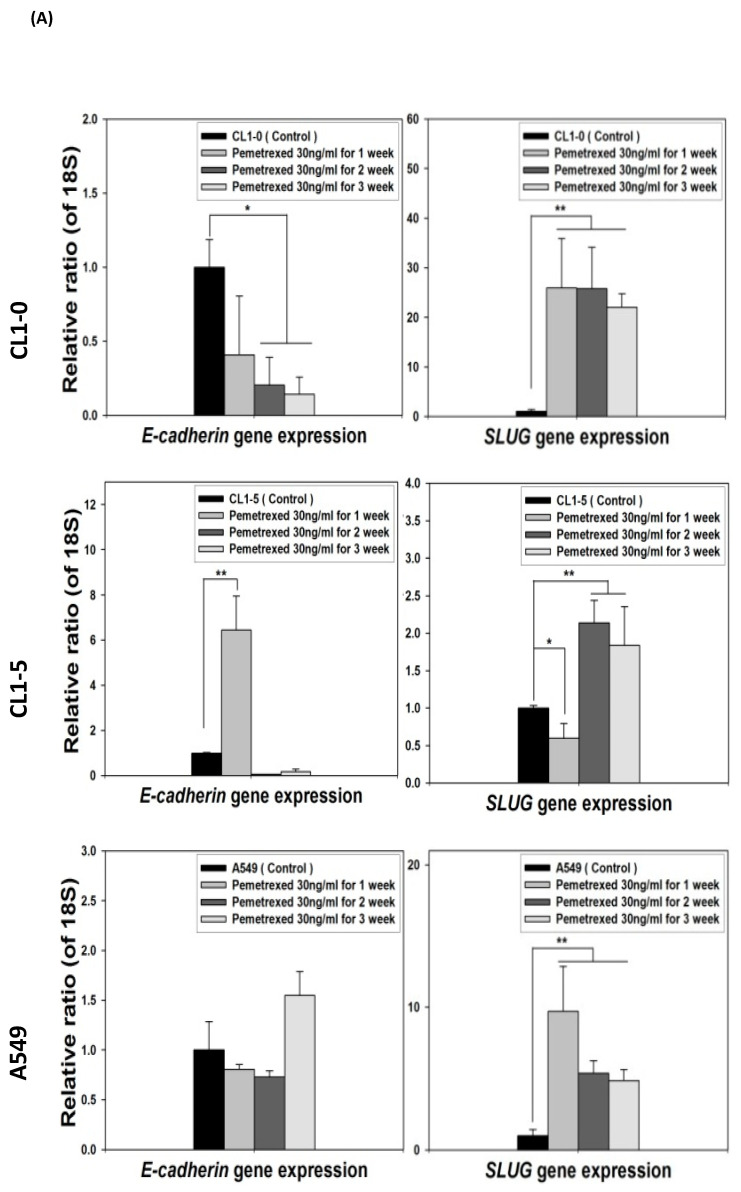
FD triggers the overexpression of SLUG transcription factors in NSCLC cells. The expression of SLUG and E-cadherin of NSCLC cell lines treated with FD or pemetrexed (30 ng/mL) were detected through RT-qPCR and Western blotting. (**A**) Gene expressions of SLUG and E-cadherin in cells treated with pemetrexed (30 ng/mL) for 1, 2, or 3 weeks increase. (**B**) Gene expressions of SLUG and E-cadherin in cells undergoing FD treatment for 2 or 4 weeks increase. In parallel, (**C**) nuclear translocation of SLUG increased detected through Western blotting. * *p* < 0.05; ** *p* < 0.01.

**Figure 5 ijms-22-08838-f005:**
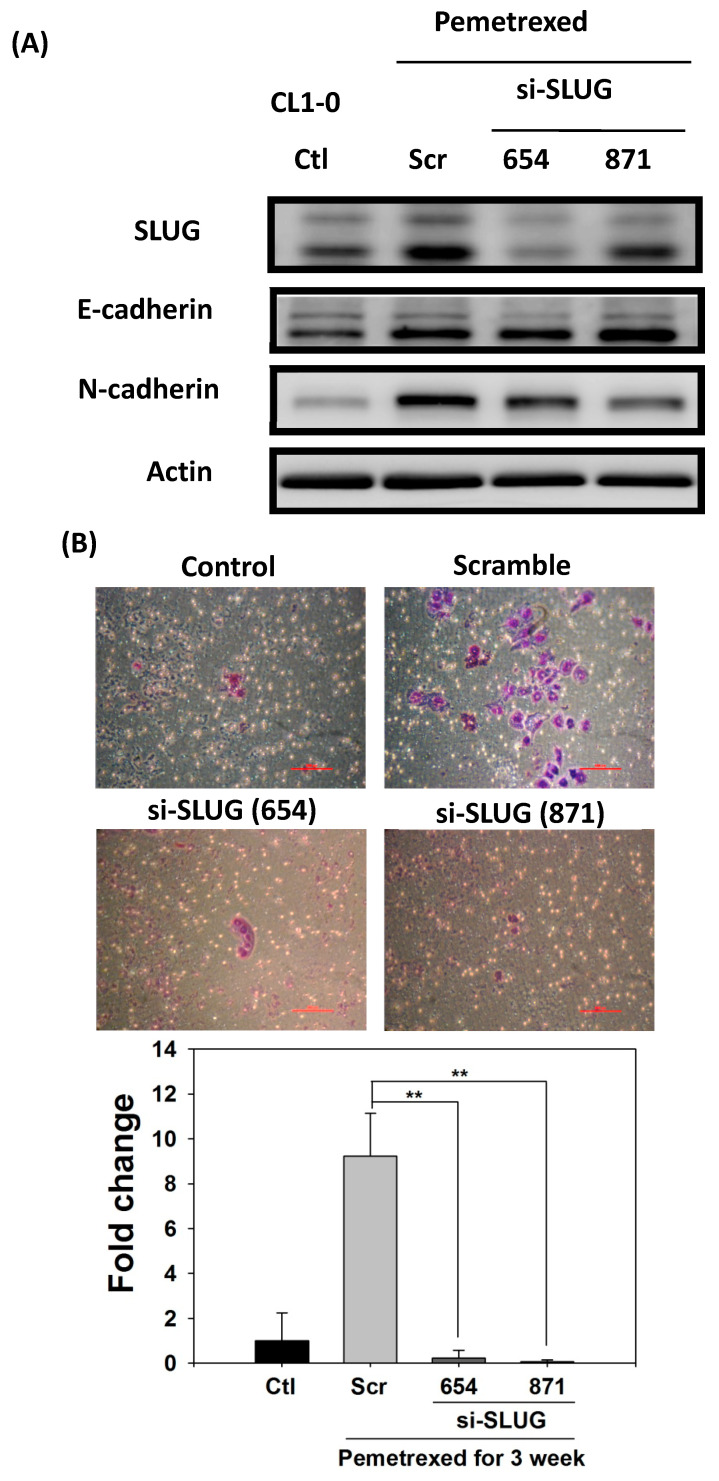
SLUG silencing suppresses invasiveness and reduces erlotinib resistance acquisition in CL1-0 cells. (**A**) Western blotting indicated that si-SLUG successfully silenced SLUG expression and could mitigate N-cadherin and SLUG expression in CL1-0 cells. (**B**) The invasiveness of cells treated with pemetrexed (30 ng/mL) was severely downregulated by SLUG silencing. The purple color represents invasive cells after the application of Liu’s stain. Scale bar (red): 100 μm. (**C**) MMP1, MMP2, MMP7, and MMP9 gene expressions significantly decreased by SLUG silencing. Gene expression was determined through RT-qPCR in untreated CL1-0 cells (control) or pemetrexed-treated (30 ng/mL) CL1-0 cells, including scramble, si-SLUG 654, and si-SLUG 871. (**D**) Gelatin zymography showing that silencing SLUG expression reduces pemetrexed-induced gelatinase activity. (**E**) SLUG silencing abolished the erlotinib resistance of CL1-0 cells acquired from pemetrexed (30 ng/mL) treatment. Control (Ctl) cells were untreated CL1-0 cells. Scramble (Scr) and si-SLUG CL1-0 cells were treated with pemetrexed (30 ng/mL) for 3 weeks. * *p* < 0.05; ** *p* < 0.01.

**Figure 6 ijms-22-08838-f006:**
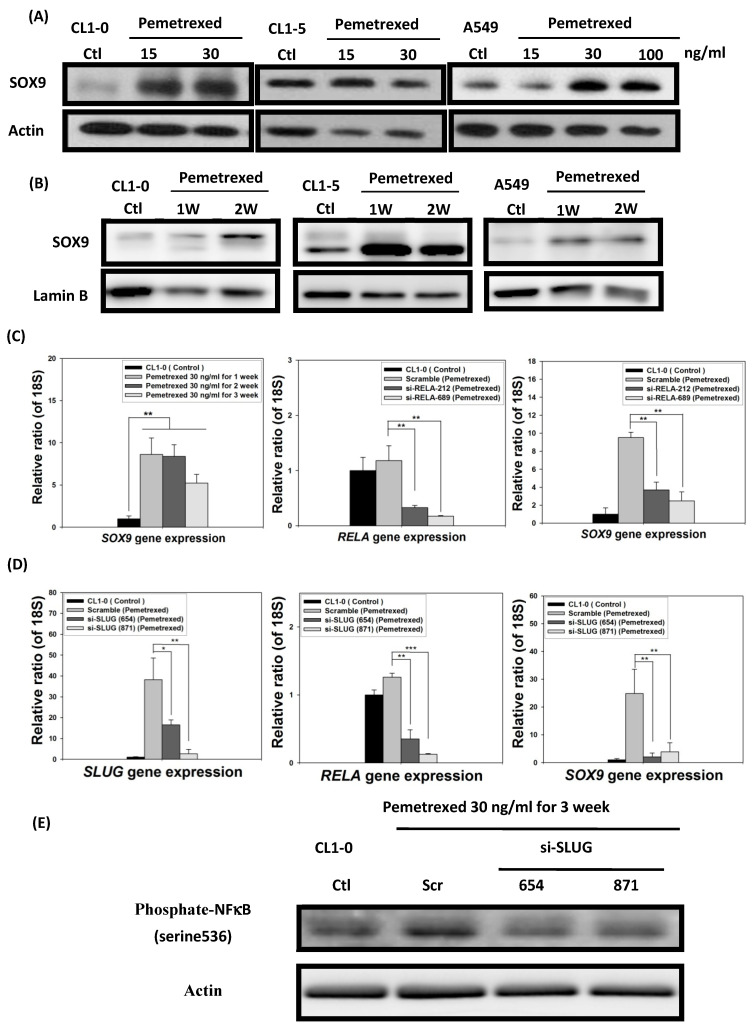
Folate antagonist unilaterally triggers the overexpression of *Sox9* transcription factors in NSCLC cells. The expression and nuclear translocation of Sox9 were analyzed through Western blotting and RT-qPCR. The (**A**) total protein expression and (**B**) nuclear translocation of Sox9 were determined through Western blotting. (**C**) NF-κB (RELA) knockdown by shRNA (si-RELA-212 and si-RELA-689) reduced Sox9 gene expression. (**D**) SLUG knockdown by shRNA (si-SLUG-654 and si-SLUG-871) also reduced NF-κB (RELA) and Sox9 gene expressions. (**E**) SLUG knockdown abolished NF-κB phosphorylation, as indicated by Western blotting, in CL1-0 cells. * *p* < 0.05; ** *p* < 0.01; *** *p* < 0.001.

**Figure 7 ijms-22-08838-f007:**
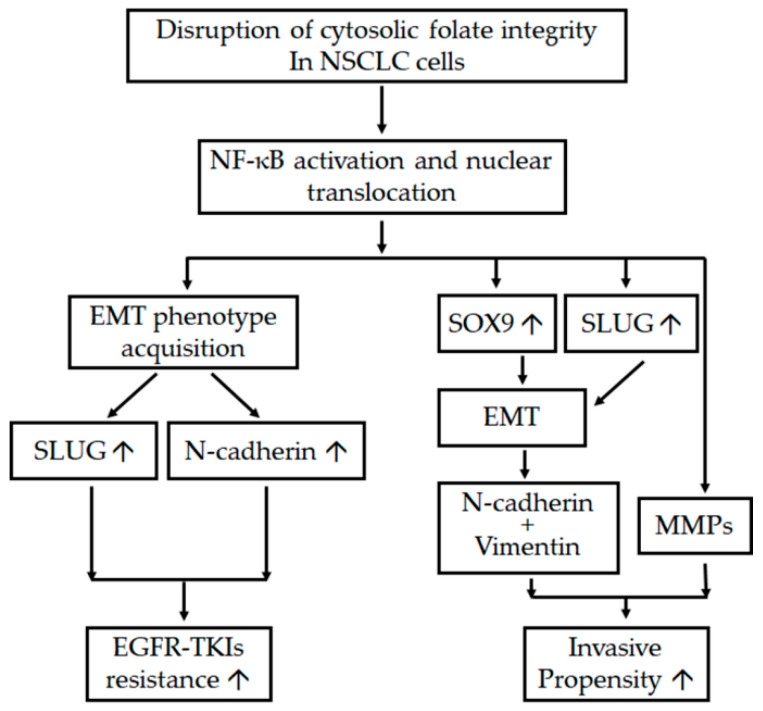
Schema of adverse effects after the disruption of folate metabolism. The diagram indicates the interaction between various mechanisms and pathways that promote the metastatic tendency and enhanced resistance of EGFR-TKIs after the disruption of folic acid metabolism in NSCLC cells. ↑ denotes upregulation.

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
