# Peer review of "Disruption of Cytosolic Folate Integrity Aggravates Resistance to Epidermal Growth Factor Receptor Tyrosine Kinase Inhibitors and Modulates Metastatic Properties in Non-Small-Cell Lung Cancer Cells"

_ijms, 2021, doi:10.3390/ijms22168838_

Round 1
Reviewer 1 Report
In this study, the authors examined the impact of disrupted folate metabolism on the metastasis capacity and acquired drug resistance in NSCLC cells. It was found that cytosolic folate deficiency (FD) induced by the use of folate deficient culture media or pemetrexed for 1 – 3 weeks enhanced the metastatic properties of NSCLC cells through upregulation of N-cadherin, NF-kB and SLUG. SLUG knockdown prevented SLUG/NF-κB/SOX9-mediated invasiveness and erlotinib resistance acquisition and suppressed pemetrexed-induced MMP expression and activity. The finding of this study suggests that the folic acid status of patients with NSCLC under treatment can considerably influence their prognosis.
- Line 85-86: "The disruption of folate integrity induced E-cadherin downregulation in CL1-0 and CL1-5 cells but not in A549 cells". The result of E-cadherin expression should be shown in Figure 1.
- Figure 1B: Samples of CL1-0 and A549 cells cultured under FD conditions were collected in Week 3 and 4, while those of CL1-5 were collected in Week 2 and 3. It was mentioned in the abstract (Line 24- 26) that "We examined whether cytosolic folate metabolism in NSCLC cells was disrupted by FD or the folate metabolism blocker pemetrexed for 1-3 weeks". Treatment duration could be deciding factor in the experiment and need to be documented accurately. What was the reason that FD culture duration for CL1-5 cells was different from CL1-0 and A549?
- Figure 1B: Since it seemed that the E-Cadherin expression was measured in cells cultured under FD condition, the WB results could be added in Figure 1B. Did the authors check Vimentin protein expression in the NSCLC cells under FD culture conditions? It should include all the proteins of interest (i.e., E-Cadherin, N-Cadherin, Vimentin, NFkB) for each treatment condition in this experiment so that the readers could see how the expression of individual proteins associated with one another.
- Figure 2: It should be clearly indicated either in the Method or in the figure legend that how long the Matrigel invasion experiment last. Also this experiment should have been done in parallel with the inserts without Matrigel coating in order to rule out the variations in the number of cells loaded to the insert and the viability and the migration ability of the cells.
- What was the reason that the Pemetrexed treatment duration in the invasion experiment was different from that in the experiment shown in Figure 1D?
- Line 104 -109: "After FD or pemetrexed treatment, matrix metalloproteinases (MMPs) 1, 2, 7, and 9 were induced in all cell lines…(Figures 2D, 2E and 2F)". Where were the results for A549 cells in Figure 2D-2F?
- Figure 2D and 2E. Changes in some of the MMP gene expression in either CL1-0 and CL1-5 did not show clear trend. Was it related to certain experimental procedures, such as culture medium replacement, cell subculture, etc.? Pemetrexed is usually stable at room temperature for 2 days. How often was the? How many hours after the last replacement of pemetrexed-containing culture medium were the cells collected for the MMP gene analysis?
- How did the pemetrexed treatment affect the doubling time of CL1-0 and CL1-5 cells? Sometimes slow proliferating cells may appear to be resistant to treatment as compared with rapid proliferating cells.
- Inconsistent results in Figure 4C and Figure 5A. As shown in Figure 4C, SLUG expression in CL1-0 was undetectable with Western blotting. However, in Figure 5A SLUG expression in CL1-0 was detectable.
- In Figure 5, was the CL1-0 control the scramble control or parental control? Cells with scramble siRNA treatment and without pemetrexed treatment should be used as the negative control.
Author Response
Answer to Reviewer 1 comments:
- Line 85-86: "The disruption of folate integrity induced E-cadherin downregulation in CL1-0 and CL1-5 cells but not in A549 cells". The result of E-cadherin expression should be shown in Figure 1.
Thank you for your suggestion. We will put this data in supplementary data Fig 1B.
- Figure 1B: Samples of CL1-0 and A549 cells cultured under FD conditions were collected in Week 3 and 4, while those of CL1-5 were collected in Week 2 and 3. It was mentioned in the abstract (Line 24- 26) that "We examined whether cytosolic folate metabolism in NSCLC cells was disrupted by FD or the folate metabolism blocker pemetrexed for 1-3 weeks". Treatment duration could be deciding factor in the experiment and need to be documented accurately. What was the reason that FD culture duration for CL1-5 cells was different from CL1-0 and A549?
Thank you for addressing this point. In fact, all three cell lines have been cultured under FD for four weeks in this study. According to our experimental observations, three cell lines would respond to treatment differently. The response of CL1-5 is rapid than both CL1-0 and A549 cell lines. Therefore, protein expression of N-cadherin and vimentin showed significant change during week 2 and week 3 for CL1-5. But it will take 3 to 4 weeks for both CL1-0 and A549 cells. The reason might be that, at original condition, CL1-5 shows more mesenchymal-like morphology than CL1-0 and A549 cells (Fig 1A).
- Figure 1B: Since it seemed that the E-Cadherin expression was measured in cells cultured under FD condition, the WB results could be added in Figure 1B. Did the authors check Vimentin protein expression in the NSCLC cells under FD culture conditions? It should include all the proteins of interest (i.e., E-Cadherin, N-Cadherin, Vimentin, NFkB) for each treatment condition in this experiment so that the readers could see how the expression of individual proteins associated with one another.
Thanks for your advice. In fact, we started with A549 cells and observed the changes of E-cadherin, N-cadherin, and vimentin after 1-4 weeks in the FD culture environment (Supplementary Figure 1A). The results found that under FD conditions, except N-cadherin showed significant change, neither E-cadherin nor vimentin showed significant changes. Therefore, based on these preliminary results, we only further examine the trend of N-cadherin in CL1-0 and CL1-5 cells under FD conditions.
- Figure 2: It should be clearly indicated either in the Method or in the figure legend that how long the Matrigel invasion experiment last. Also this experiment should have been done in parallel with the inserts without Matrigel coating in order to rule out the variations in the number of cells loaded to the insert and the viability and the migration ability of the cells.
Thank you for your suggestion. The time of the Matrigel invasion experiment is 16-18 hours. We have followed your suggestion added a detailed description in the Material and Method section (line 338-342). Since the main purpose of this study is to investigate whether the cell invasion ability has changed after the folic disruption treatment, we only performed the Matrigel invasion test but without the migration assay.
- What was the reason that the Pemetrexed treatment duration in the invasion experiment was different from that in the experiment shown in Figure 1D?
Thank you for addressing this point. The data showed in Figure 1D is the NF-kB upregulation after 1-2 weeks of folate disruption. Although NF-kB may involve EMT transformation in NSCLC cells, it might not be totally responsible for the invasion ability change after Pemetrexed treatment. According to our preliminary data, CL1-0 and CL1-5 showed a significant invasion ability change after 2 weeks of Pemetrexed treatment but not for A549 cells (Fig 2B). Only after 3 weeks of Pemetrexed treatment, A549 showed significant invasion ability change. Therefore, all the invasion experiments were performed using the cells after folate was disrupted for 2 and 3 weeks for the experiment consistency.
- Line 104 -109: "After FD or pemetrexed treatment, matrix metalloproteinases (MMPs) 1, 2, 7, and 9 were induced in all cell lines (Figures 2D, 2E and 2F)". Where were the results for A549 cells in Figure 2D-2F?
In this study, only the mRNA expression profile of MMP-1, -2, -7, and -9 for A549 under different time FD treatments have been evaluated (supplement Figure 1C). We are sorry for the non-proper description in the result section and revised it in the text.
- Figure 2D and 2E. Changes in some of the MMP gene expression in either CL1-0 and CL1-5 did not show clear trend. Was it related to certain experimental procedures, such as culture medium replacement, cell subculture, etc.? Pemetrexed is usually stable at room temperature for 2 days. How often was the? How many hours after the last replacement of pemetrexed-containing culture medium were the cells collected for the MMP gene analysis?
Although most MMPs are significantly changed in CL1-0 and CL1-5 cell lines, the gene expression of some MMPs in different cell lines has different trends. We do not know how to explain this observation. As you speculate, the reason for this unclear trend may lie in some operational problems during the experiment. Thank you for your question. We have carefully reviewed the experiment procedure while these were performed. Basically, during FD or pemetrexed treatment, the culture medium will be changed every two days and will be replaced for the last time 24 hours before the sample collection. Therefore, the difference of MMPs expression trend caused by the operation on the experiment is initially excluded.
- How did the pemetrexed treatment affect the doubling time of CL1-0 and CL1-5 cells? Sometimes slow proliferating cells may appear to be resistant to treatment as compared with rapid proliferating cells.
Thank you for mentioning this issue. In fact, as you thought, both FD and pemetrexed treatment affect the cell growth rate of all NSCLCs. Of course, drug resistance may be due to slow cell proliferation for those drugs that target cell proliferation. However, this is also the most powerful strategy for tumors to evade drug killing and wait for the chance of recurrence. This study only hopes to point out that folic disruption will strengthen the tumor survival strategy of "stop growth" and "metastasis" in NSCLCs.
- Inconsistent results in Figure 4C and Figure 5A. As shown in Figure 4C, SLUG expression in CL1-0 was undetectable with Western blotting. However, in Figure 5A SLUG expression in CL1-0 was detectable.
Thank you for addressing this point. Basically, SLUG will stay at cytosol when inactive and translocate into the nucleus when activated. The western blot data shown in Figure 4C was to detect those SLUG translocated into the nucleus. Obviously, SLUG did not activate at normal culture conditions. Therefore, it is reasonable that nuclear-translocated SLUG is almost undetectable. In contrast, the western blot of Figure 5A was to detect the total SLUG protein of the cell lysate. It is reasonable that the signal is higher than those in Figure 4C.
- In Figure 5, was the CL1-0 control the scramble control or parental control? Cells with scramble siRNA treatment and without pemetrexed treatment should be used as the negative control.
Thank you for the question. The CL1-0 control of Figure 5 including both without pemetrexed treated parental cell and pemetrexed treated cell with scramble siRNA treatment.

Reviewer 2 Report
Disruption of Cytosolic Folate Integrity Aggravates Resistance to Epidermal Growth Factor Receptor Tyrosine Kinase Inhibitors and Modulates Metastatic Properties in Non–Small-Cell Lung Cancer Cells by Shen et al is interesting, However it needs some experimental evidences to support the hypothesis.
Reviewer comments:
- The study is interesting and the hypothesis wholly depends on in vitro It is essential to validate these findings in at-least one in vivo (xenograft) model.
- The authors keep saying gene expression in many figures, it misleads. Gene expression includes both mRNA and protein. Simply say mRNA expression or protein. When you measure both mRNA and protein, it is nice to say gene expression.
- Fig 6E . Authors claim that slug knockdown abolishes NF-κB-phosphorylation, however it is partial. It nice to show by replacement blot
Author Response
Reviewer comments:
- The study is interesting and the hypothesis wholly depends on in vitro It is essential to validate these findings in at-least one in vivo (xenograft) model.
We totally agree that if there is an animal model will be better. We will plan the feasibility of the animal experiment arrangement in future studies. Highly appreciate your suggestion
- The authors keep saying gene expression in many figures, it misleads. Gene expression includes both mRNA and protein. Simply say mRNA expression or protein. When you measure both mRNA and protein, it is nice to say gene expression.
Thank you for your suggestion. We will search them through the whole article and further improve the wording.
- Fig 6E. Authors claim that slug knockdown abolishes NF-κB-phosphorylation, however it is partial. It nice to show by replacement blot
Thanks for your comment. Indeed, in Figure 6E, Slug knockdown partially affected the phosphorylation of NF-kB. It may be the reason that Slug is downstream in the signal transduction pathway (Figure 7). Therefore, slug knockdown can only affect the mRNA expression of RELA .

Round 2
Reviewer 1 Report
The authors have addressed all my concerns.
Reviewer 2 Report
In vivo data will certainly will improve the quality of the manuscript.
This manuscript is a resubmission of an earlier submission. The following is a list of the peer review reports and author responses from that submission.
Round 1
Reviewer 1 Report
[Overall]
In this manuscript titled “Disrupting of cytosolic folate integrity modulates metastatic pathway to aggravate EGFR tyrosine kinase inhibitors resistance in non-small-cell lung cancer cells”, the authors show how acid folic deprivation or its blockade affects epithelial-to-mesenchymal transition (EMT). While it is interesting to know which factors promote tumor plasticity through EMT, the finding may not offer clinical relevance for NSCLC patients. For example, clinical relevance of EMT in drug responses is well-appreciated and studied in cell lines harboring EGFR mutants. Therefore, not studying the relevance acid folic deprivation or the pathway blockade in EGFR TKI resistance with EMT is a missed opportunity. A549 cells harbor Kras and LKB1 mutations and the NSCLC patients with the genetic background will never receive EGFR TKI treatment. In vitro IC50s for NSCLC cell lines harboring EGFR kinase domain mutations are usually below 50nM whereas IC50s for cell lines with KRAS well above 1uM. The authors should know that clinically achievable serum concentration of gefitinib is around 1uM and osimertinib/afatinib ~500nM. Also, multiple labs have demonstrated that EMT in NSCLC cells are driven by multiple EMT-TFs and highly lineage specific. Thus, the use of multiple cell lines to draw conclusion is absolutely necessary.
[Comments-Major]
- Acquisition of mesenchymal phenotype is evaluated in Fig. 1B-C but vimentin is not shown in all western blot. The authors are also asked to include E-cadherin in the sets of blots since it is almost customary to show a switch from E to N-cadherin when EMT is activated. Same goes to Fig.4A.
- In Figure 2A-C, the frequency of invasion is greatest in A549 cells. However, the changes in MMPs expression or their activities were not shown for the cell line.
- The concentration of gefitinib and erlotinib used in Fig. 3 is so high that this reviewer wonders if the changes observed in IC50 are due to a cytostatic or cytotoxic effect. Annexin-V assays or immunoblots using apoptosis markers including cleaved PARP /caspase 3 may be instructive.
- The authors are asked to see if adding folate would rescue the TKI resistance seen in FD or after pemetrexed treatment.
- Vimentin should be included in Fig. 5A to demonstrate the effect on EMT phenotype after SLUG repression. Also, similar to Fig. 3 annexin V assays or cl-PARP/caspase Western blots would help to confirm an increase in TKI sensitivity after SLUG repression.
- Considering the experiments detailed in this manuscript have been done in vitro, the use of the word “metastatic” in relation to the effect of FD should be rephrased as “invasive”. The author failed to
- The manuscript should be thoroughly revised to improve the writing and spelling because there are a few errors and typos.
[Comments-Minor]
- The quality of some Western blots needs to be improved: Fig. 4D and 5B middle panels.
Reviewer 2 Report
General comments:
The submitted article contains some very interesting results but is really poorly written. The level of English will have to be significantly improved in order to publish the manuscript.
Please explain all acronyms the first time you use them.
Specific comments:
Abstract
language should be improved to make it more crisp and attention-grabbing. Interesting results are presented in a rather boring way with many language mistakes i.e.:
Line 29/30/31 – sentences poorly written
Line 33- cytotoxicity, not cytotoxic
38 – to summarize, not summarize
Introduction
Lines 48-52 – please re-write to make it more clear that what the authors mean is actually chemotherapy and targeted treatment instead of just stating the names of agents.
Line 54 – what do the authors mean by “problematic resistance”?
Line 57 – dot missing
Paragraph one – may be worth to mention other types of targeted treatments (ALK inhibitors and ROS-1 inhibitors) as well as anti-PD-1 therapy in just 1 extra sentence to broaden the view of the current treatment options for NSCLC
Paragraph two – please do not use future tense to describe results. The whole paragraph is very difficult to read and understand due to many language mistakes. It has to be re-written.
Line 68 – chemoresistance
Paragraph three – again, poorly written, with phrases like “previous evidence” or initiated factor”. Has to be re-written
I am missing a link between EGFR inhibition and folate deprivation. There is one sentence linking those two in the abstract but not in the introduction. Please describe it more thoroughly.
Results
2.1. Disruption of cytosolic folate integrity induces EMT transformation in NSCLC cells.
I think it is crucial to state at some point that Pemetrexet is a common chemotherapeutic agent used in NSCLC, this will add clinical relevance to the paper.
Line 82 – informal language (“we want to know”)
Line 84 – what metabolic defect? Please specify
Line 87 – why this concentration of folate antagonist? Did the authors establish Gi50?
Lines 90-91 – please soften the language, you cannot make a judgment that cells underwent EMT based on a picture only
Line 93 – “Along the same vein” – please re-phrase
Lines 93-95 – NFκB is involved in many pathways so the authors cannot say “folate metabolism disruption shown to be mediated by the upregulation of NF-κB”, they can only say that they observed NFκB upregulation
Figure 1 –
Title – why do the authors use the word “unilaterally”?
A – please specify when was this picture taken, at 1 or 3 weeks?
2.2. Disruption of cytosolic folate integrity promotes increased invasiveness of NSCLC cells via upregulating of MMP activities
Please correct the mistake in the title
Line 106 – first sentence makes no sense. What do authors have in mind?
Line 110 – what is Liu’s stain? Please provide reference or describe the method. What is PET?
111- what does normal mean?
113 – be observed
114 – cell lines, not cells
Lines 115-117 – the sentence is really unclear, please re-write it. If the experiment was done for A549 but the results were insignificant, please state that.
Line 117 – please do not use words like “excellently” in a scientific publication
Figure 2
A – please describe better what do we see here, what are the purple cells?
B, C – please add some short description on the actual figure as one has to read the legend to figure out what are those graphs showing
D, E – was there any reason why the authors chose to analyse RNA level instead of protein? The Figure Legend in the graph (explaining what the colour of each bar means) is too small.
2.3. Folate antagonist unilaterally enhances EGFR tyrosine kinase inhibitors (TKIs) resistance in NSCLC cells
Please provide the actual IC50 curves as a supplementary material.
Line 137 – please correct the first sentence to make it understandable
139 – not conducted, tested
147 –not resistance, resistant
152 – why unilaterally?
These are interesting results described in a very unclear manner!
Figure 3 – please re-phrase the Figure title to make it clear. Please define A and B in the Figure Legend.
2.4. Folate deprivation-mediated EGFR-TKIs resistance acquisition in NSCLC cells mechanistically linked to 161 the upregulation of SLUG and N-cadherin
Line 163-164 – again, first sentence makes no sense. Please re-write it. Same for sentence in lines 167-168
Figure 4- please correct the title
The Figure Legend in the graph (explaining what the colour of each bar means) is too small.
2.5. SLUG silencing suppresses invasive ability and alleviates erlotinib resistance acquisition of NSCLC cells.
Please clearly state that siRNA-mediated silencing was performed and explain that 2 different siRNA constructs were used.
Line 184-185 – why “subsequently”? Please re-write this sentence. Also, please remove adjectives like “dramatically” or “excellently” throughout the manuscript, they are not suitable for scientific publication.
Figure 5
A – please use appropriate language to describe your data
B – I need more explanation what is this purple colour and what is the red line at the bottom of each photo. Either in the legend or in the text. The graph under the pictures is not mentioned in the Figure Legend!
D- Figure Legend in the graph (explaining what the colour of each bar means) is too small.
E- again, I’d appreciate Gi50 curve in the supplementary
2.6. EMT in the disrupted folate metabolism mechanistically linked to the activation of the NF-κB/Sox9 200 pathway
Line 202 – please correct this sentence
Line 205 – so is it shRNA or siRNA? And what is RELA? The authors need to explain how is it connected to NFκB.
Figure 6 – Please format the title properly, again I do not understand why the authors use the word “unilaterally”
A, B – why do the authors use whole protein extract for A and nuclear fraction for B? Please explain in the text (not Figure Legend)
C, D- Figure Legend in the graph (explaining what the colour of each bar means) is too small. Also, please check whether it’s siRNA or shRNA.
Discussion
First paragraph – I sincerely ask the authors to correct all the mistakes as it’s really difficult to read.
Lines 229-230 – The authors write “Accordingly, folate deficiency had been considered a risk factor in colon cancer and liver cancer” please provide a reference for this sentence. Also, could the authors discuss this somewhat contradictory paper: StanisĹ‚awska-Sachadyn A, Borzyszkowska J, KrzemiĹ„ski M, Janowicz A, Dziadziuszko R, Jassem J, et al. (2019) Folate/homocysteine metabolism and lung cancer risk among smokers. PLoS ONE 14(4): e0214462.
Line 241 – regarding Snail, did the authors test expression of Snail or Twist? Both Snail, Slug and Twist are master regulators of EMT so it would be interesting to see the expression (at least RNA) of these other markers. If this was done but the result was insignificant, please state and discuss it.
Paragraph 3 – Sox9 is also a crucial transcription factor in stem cells (also cancer stem cells). This ties in well with the process of EMT that the authors have observed and is worth discussing.
Again, I think it is crucial to state at some point that Pemetrexet is a common chemotherapeutic agent used in NSCLC, this will add clinical relevance to the paper.
Line 273-274 – very confusing sentence
Lines 274-289 - This part could be more concise
Figure 7 – I suggest removing siRNA from the graphics to make it more clear
Materials and methods
Please move the information about antibodies from section 4.1 to section 4.3
Section 4.3 – please improve the Western blot description adding information about blocking solution, temperature of primary antibody incubation, the fact that secondary antibodies were used etc.
Section 4.4 – please add the conditions for the qPCR reaction and the amount of cDNA used
Section 4.5 – more details have to be added to this description
Line 356 – please explain what conditioned media is
Reviewer 3 Report
This is an interesting work with the aim to explore effect of folate metabolism impairment on metastasis potential, especially mediated by epithelial to mesenchymal transition (EMT) and its impact on EGFR-TKIs sensibility.
As reported by authors, folic acid deprivation is a frequent event in NSCLC and can impact on many cancer cell properties such as cell viability, invasiveness or resistance to therapies. Originality mainly relies on SLUG/Sox9 identification. However, some troubling points need to be clarified.
As major revisions, the following remarks can be reported:
- Abstract (lines 23-24) / introduction / discussion : authors addressed folate deprivation through malnutrition. Malnutrition is a very common challenge in NSCLC and can be mentioned. However, Alimta (=pemetrexed) is a major cytosolic folate disruptor and cytotoxic drug, used in daily practice for non-squamous non-small cell lung cancer. Introduction, problematic and discussion need to be more focused on this highly used molecule in the current clinical strategy and its potential impact on secondary pemetrexed-induced resistance. Finally, it would allow authors to discuss on drug strategies and especially on therapeutic sequences concerning alimta and EGFR-TKIs. This approach would make it more relevant on a translational point of view.
- Additionally, lines 54 and 74 mentioned a “resistance to EGFR-TKIs”. However, this resistance can be supported by many processes such as primary or secondary mutations, bypass signaling activations and in an EGFR-dependent and independent manners (involving EMT for example). This must be clarified in order to better set the molecular context.
- Figure 1 : Morphological and Western Blot characterization are interesting. However, EMT process need to be evaluated both on mesenchymal and epithelial features. E-cadherin characterization would be useful.
- A large part of the discussion refers to oxidative stress (ONS). Observing that any experience investigates ONS, this part could be limited to SLUG/sox9 interaction. Additionally, discussion should focus on potential impact of pemetrexed disruption in clinical strategies for wild type EGFR cancers pretreated with pemetrexed and harboring a primary or secondry resistance to TKIs. Still in discussion section, many previous works reported a strong interrelation between Pemetrexed and EGFR-TKIs. Some results could be discussed : La Monica JTO 2016 (gefinitib/pemetrexed combination with EMT consideration) ; On another part, Dae Ho Lee et.al (PMID 25672577) focusing on Pemetrexed-Erlotinib, Pemetrexed Alone, or Erlotinib Alone as second line in a wild-type EGFR population appears as a very relevant protocol to confront those preclinical data.
Finally, minor revisions can also be reported:
- Line 33: “tendency of NSCLC cells to metastasis” is not correct. Terms “Invasiveness”, “aggressiveness” or “metastatic properties” seem more adapted. Please consider it.
- Introduction line 51 – 55 : TKIs-EGFR strategy in a wild-type EGFR context need to be checked and clarified. “Third-line therapy” is incorrect and EGFR-TKIs remain an option for patient with a progression after a platinum based-regimen. For example, clinician can use immunotherapy, followed by one (or more) other chemotherapies before EGFR-TKIs.
- Figure 1 B : Could authors explain the difference for CL1-5 ? only CL1-5 cell line was treated up to 3w against 4w for the other ones.
- Figure 1 C : Only CL1-5 harbored lower level of actin. Is this artefactual ? Could authors discuss on it ?
- Figure 3 : could authors explain absence of results related on IC50 EGFR-TKIs for A549 cell line ?